# Beware of Model Collapse! Fast and Stable Test-time Adaptation for Robust Question Answering

**Yi Su[1][*], Yixin Ji[1][*], Juntao Li[1][†], Hai Ye[2], Min Zhang[1]**

[1]Institute of Computer Science and Technology, Soochow University, China
[2]Department of Computer Science, National University of Singapore
yisunlp@outlook.com; jiyixin169@gmail.com;
yehai@comp.nus.edu.sg;
{ljt,minzhang}@suda.edu.cn

## Abstract

Although pre-trained language models (PLM) have achieved great success in question answering (QA), their robustness is still insufficient to support their practical applications, especially in the face of distribution shifts. Recently, test-time adaptation (TTA) has shown great potential for solving this problem, which adapts the model to fit the test samples at test time. However, TTA sometimes causes model collapse, making almost all the model outputs incorrect, which has raised concerns about its stability and reliability. In this paper, we delve into why TTA causes model collapse and find that the imbalanced label distribution inherent in QA is the reason for it. To address this problem, we propose Anti-Collapse Fast test-time adaptation (Anti-CF), which utilizes the source model's output to regularize the update of the adapted model during test time. We further design an efficient side block to reduce its inference time. Extensive experiments on various distribution shift scenarios and pre-trained language models (e.g., XLM-RoBERTa, BLOOM) demonstrate that our method can achieve comparable or better results than previous TTA methods at a speed close to vanilla forward propagation, which is 1.8× to 4.4× speedup compared to previous TTA methods. Our code is available at https://github.com/yisunlp/Anti-CF.

## 1 Introduction

Pre-trained language models (PLMs) have achieved great success on many NLP tasks (Devlin et al., 2019; Liu et al., 2019; Lewis et al., 2020a; Raffel et al., 2020; Brown et al., 2020; OpenAI, 2022, 2023; Touvron et al., 2023). However, their success is based on the assumption that the test distribution is consistent with the training distribution. In many scenarios, this assumption is not true, such as adversarial attack (Wang et al., 2022), cross-lingual (Li et al., 2021), cross-domain (Ramponi and Plank,

---

[*]Equal Contribution.
[†]Corresponding author.

2020), and so on. This situation is known as distribution shift. Unfortunately, even the most advanced models currently available, such as ChatGPT, do not perform well under the distribution shift (Ye et al., 2023; Wang et al., 2023).

To address this problem, researchers have proposed many approaches such as adversarial training (Zhu et al., 2020; Wang et al., 2021a), data augmentation (Zhou et al., 2021; Chen et al., 2021a). These methods improve the robustness of the model by changing the training strategy, but according to the No Free Lunch Theorem (Wolpert and Macready, 1997), a fixed model still cannot perform perfectly in all distribution-shifted scenarios. Therefore, some works (Wang et al., 2021b; Sun et al., 2020; Niu et al., 2022; Ye et al., 2022) explore how to update the model during the testing phase to adapt it to the distribution shifts of the test samples, called Test Time Adaptation (TTA). A typical approach (Wang et al., 2021b) uses the Shannon entropy of the probability given by the model as the loss to update itself. However, due to the unreliable output of the model, TTA may accumulate erroneous information learned in test samples, leading to model collapse and a sharp decline in model performance, which makes TTA extremely unstable and unreliable in practical applications.

To solve this problem, we take QA task as an example and investigate why TTA causes the model collapse. Our experiments indicate that the main reason for the model collapse is the imbalanced label distribution of the test data. In contrast to the direct inference, TTA exacerbates this imbalanced distribution, making all outputs of the model to be a specific class. Therefore, we propose Anti-Collapse Fast test-time adaptation (Anti-CF), which utilizes the output of the source model as a soft label to regularize the update of the adapted model during test time to ensure that the adapted model will not deviate too far from the source model, thus avoiding model collapse.

However, to obtain the output of the source model and the adapted model, we need to keep the parameters of two models and conduct forward propagation twice, which will bring a lot of additional costs in practical applications. Therefore, we freeze the source model and add an efficient side block as the adapted model to reduce the cost of additional forward propagation and back propagation. Extensive experiments on various distribution shift scenarios and PLMs demonstrate that our method can achieve comparable or better results than previous TTA methods at a speed close to vanilla forward propagation, which is 1.8× to 4.4× speedup compared to previous TTA methods.

Overall, our contributions in this work include:

- We investigate why TTA causes model collapse in QA and find that the imbalanced label distribution inherent in QA is the reason for it.
- We propose Anti-Collapse Fast test-time adaptation (Anti-CF) to solve the problem that TTA sometimes leads to model collapse.
- Experimental results show that Anti-CF can effectively prevent the model from collapsing with a fast inference speed. It improves the stability and reliability of TTA in practical applications.

## 2 Preliminary

In this section, we begin by introducing extractive question answering and the application of TTA to enhance its robustness. Subsequently, we focus on Tent (Wang et al., 2021b) as a case to discuss the potential risks of TTA, specifically the occurrence of model collapse. Additionally, we conduct an analysis of generative QA in Section 5.5.

### 2.1 Test-time Adaptation for Question Answering

In extractive QA, the input of the model is a combination of a context and a question. The goal is to determine the start and end positions of the answer within the context, where the text between them represents the answer. However, in practice, the context-question pairs are often too long to be directly processed by the model. To address this, we divide the context into smaller spans. For each span, the model predicts the start and end positions of the answer within that specific span. In cases where the model determines that the answer does not exist within a given span, it will output the start and end positions of a special token, such as the [CLS] token, indicating the absence of an answer in that particular span.

Assume that we have training data pairs $\{x_s^i, y_s^i\}_{i=1}^{n_s} \in \mathcal{D}_s$ with the distribution $\mathcal{P}_s$, where $x_s^i \in \mathcal{X}_s$ refers to the span-question pair and $y_s^i \in \mathcal{Y}_s$ refers to the corresponding position of the answer. After successful training on the training data, we obtain a model $f_\theta : \mathcal{X}_s \rightarrow \mathcal{Y}_s$. In the test phase, we have test data samples $\{x_t^i\}_{i=1}^{n_t} \in \mathcal{X}_t$ with underlying corresponding labels $\{y_t^i\}_{i=1}^{n_t} \in \mathcal{Y}_t$. The distribution of the test data is $\mathcal{P}_t \neq \mathcal{P}_s$. If we use the trained model $f_\theta$ to predict test samples, the model's performance will degrade due to the distribution shift. Test-time adaptation provides a promising paradigm to mitigate performance degradation, which updates the model during the testing phase to accommodate the distribution shifts of test data. Tent (Wang et al., 2021b) and EATA (Niu et al., 2022) use the model's prediction probability of the test sample as a soft label to optimize the model by minimizing the entropy:

$$\mathcal{L}(x_t) = -\sum_c p\left(y_c|x_t\right) \log p\left(y_c|x_t\right) \quad (1)$$

where $p\left(y_c|x_t\right)$ is the probability of the c-th category of $x_t$.

### 2.2 The Risk of TTA Leading to Model Collapse

We use XLM-RoBERTa-base (Conneau et al., 2020) as the backbone to train a source model on SQuAD (Rajpurkar et al., 2016) and evaluate it on NaturalQA (Kwiatkowski et al., 2019), which is a cross-domain setting. We compare the results of direct inference and Tent. We experimented with various optimizers and learning rates for Tent, as illustrated in Figure 9. We find that no matter what kind of optimizer and what learning rate we set, the performance of the model will decrease. Obviously, the smaller the learning rate, the smaller the impact of TTA on the model. However, if we set the learning rate to be very small, this will make TTA almost ineffective, and if we set the learning rate to a more reasonable range, the model may collapse. TTA does not always improve the robustness of the model and even has a risk of model collapse, which would seriously hinder the application of TTA in real-world scenarios.

To explore why TTA causes model collapse, we study the entropy of test data. As Figure 2 shows, the entropy of Tent sharply decreases until approaching 0, which means the collapsed model

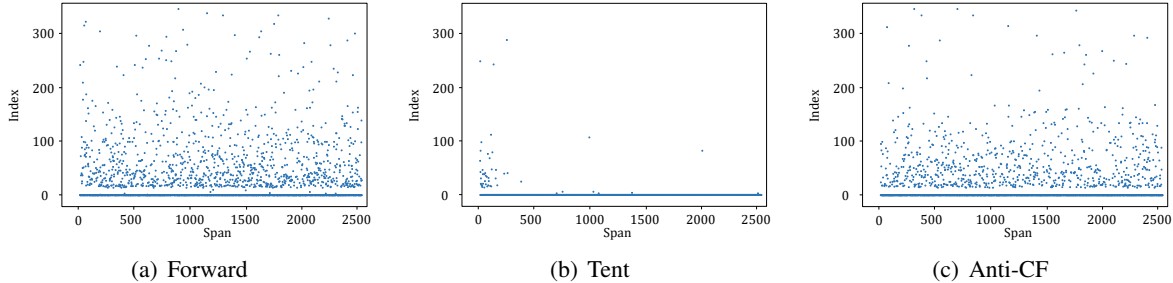

| (a) Forward | (b) Tent | (c) Anti-CF |

Figure 1: The start positions of NaturalQA given by the model using Forward, Tent, and Anti-CF. Forward: direct inference.

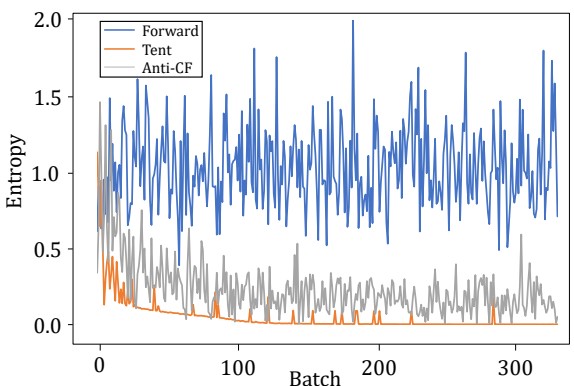

Figure 2: The entropy of NaturalQA predicted by the model using Forward, Tent, and Anti-CF. Forward: direct inference.

makes wrong predictions with high confidence. We further explored the start positions given by the model (Figure 1, the ground truth and end positions can be found in the Appendix C). We find that 0 (indicates that the answer is not in this span) accounts for the majority. Furthermore, after using Tent, the model's prediction tends to be more and more inclined towards 0, which directly leads to almost all of the model's predictions being incorrect. Therefore, the imbalanced distribution of test labels has led to TTA leaning too much towards the majority class during the update process, resulting in all outputs being biased towards some specific classes, which is why TTA causes model collapse.

## 3 Method

In this section, we propose Anti-collapse Fast Test-time adaptation (Anti-CF). Anti-CF consists of two strategies. (1) *Entropy minimization with source constraints* (Section 3.1) seeks to ensure that the adapted model does not deviate too far from the source model, thus avoiding the occurrence of model collapse. (2) *Efficient side block* (Section

3.2) aims to reduce the inference time by building a small network next to the backbone.

### 3.1 Entropy Minimization with Source Constraints

To solve the problem we mentioned in Section 2.2, we want to use the output of the source model as a constraint to the adapted model during test time so that the adapted model does not deviate too far from the source model, thus avoiding model collapse.

Like many previous TTA methods, we also choose entropy minimization as one of our optimization goals, which can be formulated as:

$$\mathcal{L}_e = -\frac{1}{n} \sum_{i=0}^{n} \sum_{c} p_a\left(y_c|x_i\right) \log p_a\left(y_c|x_i\right) \quad (2)$$

where $\{x\}_i^n$ is a batch of test samples and $p_a\left(y_c|x_i\right)$ is the prediction probability of $x_i$ given by the adapted model. We use forward Kullback-Leibler (KL) divergence to constrain the update of the adapted model, which will make the output of the adapted model close to that of the source model:

$$\mathcal{L}_c = \frac{1}{n} \sum_{i=0}^{n} \sum_{c} p_s(y_c|x_i) \log \frac{p_s(y_c|x_i)}{p_a(y_c|x_i)} \quad (3)$$

where $p_s\left(y_c|x_t\right)$ is the probability of the c-th category given by the source model. We introduced a hyper-parameter $\alpha$ to balance the two losses, so the loss function of Anti-CF is:

$$\mathcal{L} = (1-\alpha)\mathcal{L}_e + \alpha\mathcal{L}_c \quad (4)$$

We can briefly analyze why Anti-CF can avoid model collapse. Suppose model collapse has occurred, with extremely low entropy like that in Section 2.2. At this point, the first part of the loss $\mathcal{L}_e$ is close to 0, and the loss approximately has only the second part $\mathcal{L}_c$. Therefore, the main objective of the loss is to pull the adapted model closer

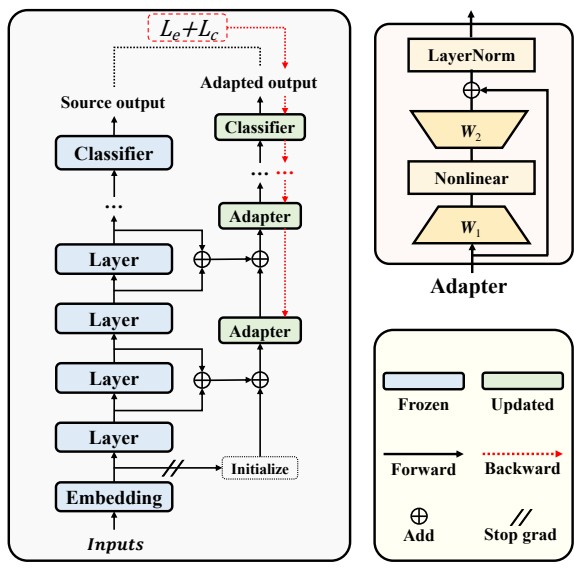

Figure 3: The structure of efficient side block. There is an adapter added between every two layers. Gradients only propagate through adapters, and the backbone is frozen. The adapted output will be used as the final output for prediction, while the source output is only used to constrain the update of the efficient side block.

to the source model, which effectively avoids the occurrence of model collapse.

## 3.2 Efficient Side Block

To minimize Eq.4, we need to obtain the predicted probability of the source and adapted model. However, this requires at least two forward propagation and one back propagation for each sample, which undoubtedly dramatically increases the cost of practical application. To break this dilemma, we propose an efficient side block, which is plugged into the backbone as the adapted model so that we only need one forward propagation to obtain the two outputs simultaneously. In addition, the gradient only back propagates through the efficient side block, reducing the cost of back propagation.

As shown in Figure 3, the efficient side block consists of a series of adapter modules (Houlsby et al., 2019). We plug an adapter between every two Transformer layers (Vaswani et al., 2017). The input of each adapter is the combination of the hidden state of its corresponding two layers and the hidden state of the previous adapter:

$$\boldsymbol{h}_k = \text{Layer}_k(\boldsymbol{h}_{k-1}), k = 2i, 2i+1 \quad (5)$$

$$\boldsymbol{s}_i = \text{Adapter}_i(\boldsymbol{s}_{i-1} + \boldsymbol{h}_{2i} + \boldsymbol{h}_{2i+1}) \quad (6)$$

where $\boldsymbol{h}_k$ is the hidden state of the $k$-th Transformer layer, $\boldsymbol{s}$ is the hidden state of the $i$-th adapter,

$i$ ranges from 1 to the number of layers in the efficient side block. Both $\boldsymbol{h}_0$ and $\boldsymbol{s}_0$ are initialized as embedding outputs. For example, the XLM-RoBERTa-large has 24 Transformer layers, we take 12 adapter modules as the side block.

When a sample is given, since the backbone and side block are parallel, only one forward propagation is needed to obtain the output of the source model and the adapted model. During back propagation, the backbone is frozen and only the parameters of the efficient side block are updated, which prevents gradient propagation in the backbone, thus significantly accelerating the back-propagation speed.

Since the efficient side block is additionally plugged into the backbone in the TTA phase, it is not trained in the training phase. Thus, its parameters are randomly initialized. We believe that the efficient side block without learning task-specific information may cause performance degradation of TTA, so we train the efficient side block before performing TTA, which we call the warmup process. Since the warm-up phase only learns task-related information, the warmup data can be either the training data of the original model or other available data of the same task.

## 4 Experiments

### 4.1 Datasets and Evaluation Metrics

To verify the effectiveness of our proposed Anti-CF, we conduct experiments in three distribution shift scenarios: adversarial attack, cross-lingual, and cross-domain. Datasets we use as the following:

**NoiseQA** (Ravichander et al., 2021) adds three common types of noise to the SQuAD dataset (Rajpurkar et al., 2018): speech recognition, keyboard inputs, and translation systems. Then, NoiseQA adds natural and synthetic noise, generating two subsets named NoiseQA-syn and NoiseQA-na.

**XQuAD** (Artetxe et al., 2020) is a cross-lingual QA dataset. XQuAD translates a subset of the development set from SQuAD into ten languages.

**MLQA** (Lewis et al., 2020b) is also a cross-lingual dataset. Compared to XQuAD, it annotations the new dataset from English Wikipedia and translates it into six languages.

**MRQA** (Fisch et al., 2019) is a cross-domain dataset which includes HotpotQA (Yang et al., 2018), NaturalQA (Kwiatkowski et al., 2019), NewsQA (Trischler et al., 2017), SearchQA (Dunn et al., 2017), and TriviaQA (Joshi et al., 2017).

| Models | NoiseQA-syn | | NoiseQA-na | | XQuAD | | MLQA | | MRQA | | Avg. | | Time |
|---|---|---|---|---|---|---|---|---|---|---|---|---|---|
| | EM | F1 | EM | F1 | EM | F1 | EM | F1 | EM | F1 | EM | F1 | |
| xlmr-base | 66.61 | 78.64 | 66.08 | 77.92 | 55.59 | 71.43 | 48.27 | 65.83 | 40.12 | 52.79 | 55.33 | 69.32 | 4.25 |
| +Tent | 68.14 | 79.47 | 67.44 | 78.78 | 56.40 | 71.74 | 48.11 | 65.61 | 20.77 | 32.35 | 52.17 | 65.59 | 9.43 |
| +EATA | 68.25 | 79.55 | 67.57 | 78.88 | 56.42 | 71.64 | 48.11 | 65.53 | 19.16 | 30.62 | 51.90 | 65.24 | 11.01 |
| +OIL | **68.36** | **79.55** | **67.74** | **79.01** | 56.69 | **71.92** | 48.28 | 65.62 | 37.65 | 49.60 | 55.74 | 69.14 | 23.07 |
| +SAR | 67.63 | 79.30 | 67.36 | 78.76 | 55.94 | 71.77 | 48.01 | 65.41 | 31.71 | 42.68 | 54.13 | 67.58 | 18.30 |
| +Ours | 68.08 | 79.45 | 67.50 | 78.82 | **56.75** | 71.72 | **48.32** | 65.66 | **40.16** | **52.86** | **56.16** | **69.70** | **5.23** |
| xTune | 70.92 | 81.49 | 69.72 | 80.65 | 58.81 | 73.77 | 50.66 | 67.86 | 43.36 | 56.04 | 58.70 | 71.96 | 4.25 |
| +Tent | **71.52** | 81.89 | 70.78 | **81.24** | 59.48 | 74.08 | 51.10 | 68.07 | 26.96 | 38.24 | 55.97 | 68.70 | 9.43 |
| +EATA | 71.49 | **82.01** | 70.67 | 81.14 | 59.36 | 73.91 | 51.30 | 68.24 | 24.51 | 35.70 | 55.47 | 68.20 | 11.01 |
| +OIL | 71.42 | 81.91 | **70.80** | 81.15 | **59.66** | **74.16** | **51.36** | 68.28 | 39.54 | 51.63 | 58.57 | 71.42 | 23.07 |
| +SAR | 70.90 | 81.46 | 70.34 | 80.85 | 59.23 | 73.93 | 51.19 | 68.11 | 36.73 | 48.50 | 57.68 | 70.57 | 18.30 |
| +Ours | 71.32 | 81.75 | 70.19 | 80.91 | 59.01 | 73.83 | 51.18 | **68.28** | **43.44** | **56.05** | **59.02** | **72.16** | **5.23** |
| xlmr-large | 65.55 | 79.91 | 64.17 | 78.37 | 63.15 | 78.77 | 54.40 | 72.50 | 46.17 | 59.45 | 58.69 | 73.80 | 12.37 |
| +Tent | 70.07 | 82.74 | 67.21 | 80.34 | 63.66 | 79.02 | 54.50 | 72.51 | 27.65 | 36.83 | 56.62 | 70.29 | 26.27 |
| +EATA | 70.59 | 83.03 | 68.37 | 81.07 | 63.65 | 78.97 | 54.60 | 72.52 | 24.87 | 33.43 | 56.42 | 69.81 | 29.94 |
| +OIL | 69.83 | 82.92 | 68.22 | 81.05 | **63.86** | **79.18** | 54.77 | 72.72 | 40.98 | 52.29 | 59.53 | 73.63 | 66.37 |
| +SAR | 69.29 | 82.38 | 67.70 | 80.56 | 63.52 | 78.98 | 54.43 | 72.37 | **46.41** | 59.25 | 60.27 | 74.71 | 52.47 |
| +Ours | **71.53** | **83.27** | **69.28** | **81.08** | 63.78 | 79.00 | **54.93** | **72.81** | 46.24 | **59.47** | **61.15** | **75.13** | **15.36** |

Table 1: Main results (%). We select three source models: XLM-RoBERTa-base (xlmr-base), xTune-XLM-RoBERTa-base (xTune), and XLM-RoBERTa-large (xlmr-large). **Bold**: the best results.

We report the **EM** (Exact Match) and **F1** score for each dataset. We also provide the **Time** of each TTA method, which means the time (ms) the model requires to process a sample.

### 4.2 Baselines

We use the following strong baselines as a comparison to verify the effectiveness of Anti-CF.
**Tent** (Wang et al., 2021b) updates the model by minimizing the entropy of test samples.
**EATA** (Niu et al., 2022) only updates samples with low entropy to improve the reliability of pseudo-labels. In addition, it restricts the updating of the model from changing too much to avoid forgetting.
**OIL** (Ye et al., 2022) uses a teacher-student paradigm to increase the reliability of the pseudo-labels. It utilizes the output of the teacher as the pseudo-labels to guide the update of the student.
**SAR** (Niu et al., 2023) utilizes the sharpness-aware optimizer to enhance the reliability of the model updating process and enable adaptation to highly challenging environments.

### 4.3 Implementation Details

In our main experiments, we utilize the XLM-RoBERTa-base/large as the backbone model. In addition, we use xTune (Zheng et al., 2021), a strong robustness tuning method to train a source model on XLM-RoBERTa-base. We train the source

model on SQuAD using the default training setup from XTREME (Hu et al., 2020).

For all baselines, to speed up TTA as much as possible, we follow the setup of Su et al. (2023) and only tune all LayerNorm parameters. When reproducing EATA, we discard the step of filtering redundant samples following (Niu et al., 2022) because this method is unsuitable for NLP data.

For Anti-CF, we set the adapter's hidden size the same as the source model's hidden size. Unlike the setting in OIL, we believe that TTA should not select a set of hyper-parameters for each test set individually because in a complex and variable real-world scenario, we cannot make a careful hyper-parameter selection for each distribution shift.

We run all experiments with different random seeds three times and take the averaged result as the final experimental results. We tune the model with the learning rate in {5e-5, 1e-4, 5e-4} and set the batch size as 8. We use the validation set of SQuAD to warmup the efficient side block for one epoch with the learning rate of 5e-4. All experiments are completed on NVIDIA RTX 3090 GPU. Details of all hyper-parameters are given in Appendix B.

### 4.4 Main Results

Table 1 shows the main results of each TTA method. We have the following observations:
**Anti-CF can effectively avoid model collapse.** On

MRQA, Tent and EATA cause the model collapse, resulting in a significant decrease in performance compared to the source model, with the highest EM decreasing by 21.3% (46.17% → 24.87%). Although OIL can alleviate model collapse with imitation learning from the mean teacher, there will still be significant performance degradation, at most from 46.17% to 40.98% on EM. Even the latest baseline SAR cannot completely avoid model collapse. However, Anti-CF has no performance degradation on any dataset, avoiding the model collapse that other TTA methods may encounter. We also plot Anti-CF's start position distribution (Figure 1(c)) and entropy of NatrualQA (Figure 2). We note that the entropy on Anti-CF decreases slowly compared to Tent, and the predictions avoid completely lean towards the majority of the labels. This corroborates that Anti-CF can effectively prevent the model collapse caused by TTA.

**Anti-CF has superior inference speed among all TTA methods.** Anti-CF is about 1.8 times faster than Tent, two times faster than EATA, 4.4 times faster than OIL, 3.4 times faster than SAR, and only about 20% slower than vanilla forward. This speed is faster than all existing TTA methods and has a vast advantage in real-world applications.

**Anti-CF can achieve comparable or better results than other TTA methods.** On the NoiseQA, XQuAD, and MLQA datasets, each TTA method performs well and can achieve performance improvements based on the source model. Anti-CF can achieve comparable or better results than other TTA methods without model collapse. Among them, when using xlmr-large as the source model, the EM of Anti-CF is 5.98% higher than that of vanilla forward and 0.94% higher than the best performance among other TTA methods on NoiseQA-syn. On average, Anti-CF has a stable improvement effect on all source models.

**Anti-CF has great potential in real-world applications.** Previously, TTA was challenging to apply in real-world applications due to its instability. Although it achieves performance improvements, it will also sometimes causes the model to collapse, resulting in almost all output being false, which is unacceptable in real-world applications. Anti-CF can avoid it. In addition, many existing TTA methods are becoming increasingly complex, incorporating technologies such as contrastive learning (Chen et al., 2022), data augmentation (Liu et al., 2021; Zhang et al., 2022), and knowledge distilla-

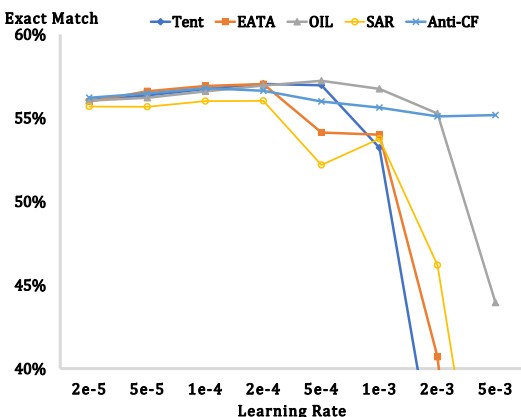

Figure 4: Effect of learning rate.

tion (Ye et al., 2022), resulting in a much slower inference speed than vanilla forward, increasing its cost in real-world applications. The inference speed of Anti-CF is close to vanilla forward, which can meet the speed needs of practical applications.

## 5 Further Analysis

### 5.1 Effects of the Learning Rate

The learning rate is a very important hyper-parameter of TTA. Choi et al. (2022) shows that a large learning rate may cause the model to collapse, while a small learning rate can make TTA almost ineffective. However, careful hyper-parameter selection for TTA during test time is not feasible in practice. Therefore, an advanced TTA approach should be less sensitive to the learning rate. We use the XLM-RoBERTa-base as the source model to test the sensitivity of each TTA method to the learning rate on the XQuAD dataset. The result is shown in Figure 4. We can observe that Tent, EATA and SAR are very sensitive to the learning rate. With the increase in the learning rate, the performance of them drops rapidly after reaching the maximum, which indicates that they are prone to model collapse under a large learning rate. OIL performs better than Tent, EATA and SAR, but it still rapidly deteriorates after maintaining performance for a while. In contrast, Anti-CF is less sensitive to the learning rate. As the learning rate increases, the performance of Anti-CF will slowly decline until it approaches the performance of the source model.

### 5.2 Effects of $\alpha$

$\alpha$ is an important hyper-parameter of Anti-CF, significantly influencing the results. To thoroughly investigate its impact, we conduct experiments on the NaturalQA dataset. Figure 5 shows the effects of $\alpha$.

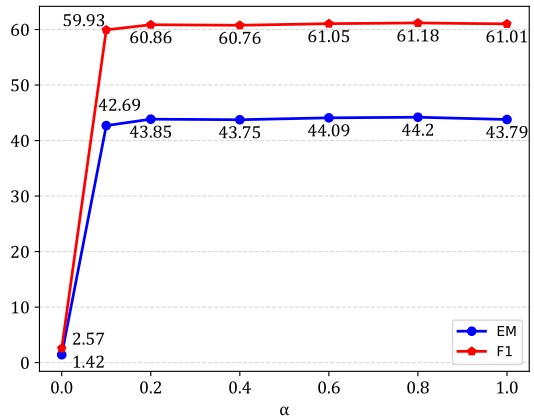

Figure 5: EM and F1 of Anti-CF on NaturalQA under different $\alpha$. We use xlmr-base as our source model and 5e-4 as the learning rate.

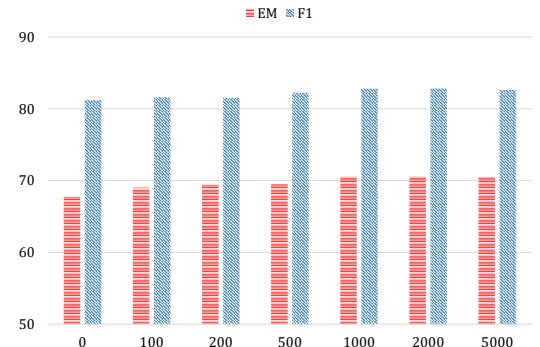

Figure 6: EM and F1 of Anti-CF under different quantities of warmup data.

When $\alpha$ is set to 0, indicating that Anti-CF does not impose any constraints on the adapted model, the model quickly collapses. However, even a slight increase in $\alpha$, such as 0.1, provides enough constraint to prevent the model from collapsing, resulting in a remarkable improvement in the EM score from 1.42% to 42.69%. This change demonstrates the effectiveness of Anti-CF.

### 5.3 Effects of Warmup Data

We explore the impact of the amount of warmup data on Anti-CF. We use xlmr-large as the source model on the NoiseQA-syn dataset and conducted experiments with different amounts of warmup data. We randomly sample the warmup data from the validation set of SQuAD. As shown in Figure 6, even if we do not perform warmup, Anti-CF still achieves performance over direct inference, which may be because source constraints make the efficient side block learn the knowledge from the source model in the process of TTA. As the amount of warmup data grows, Anti-CF performs better,

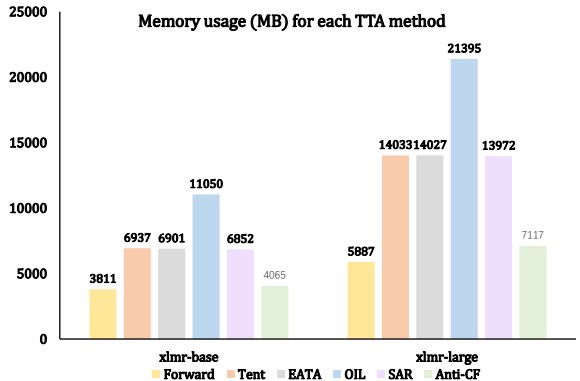

Figure 7: Memory usage of various TTA methods.

and when the amount of warmup data reaches the thousand level, the performance of Anti-CF reaches a plateau. It indicates that warmup is essential to harness the power of Anti-CF and is data-efficient, requiring only hundreds or thousands of samples to achieve exciting performance gains when data resources are limited. We speculate that because the model capacity of the efficient side block is small, there is less demand for training data.

### 5.4 Memory Usage for TTA Methods.

In practical applications, TTA requires additional memory, which poses challenges when deploying on lightweight devices with limited memory resources. We discover that the efficient side block of Anti-CF can potentially solve this problem by reducing the memory required for back propagation. To demonstrate this, we record the memory required by each TTA method in Figure 7. It is evident that Anti-CF incurs minimal additional memory compared to vanilla forward. This is because Anti-CF does not require passing through the backbone during back propagation and does not need to record gradients within it. In contrast, Tent and EATA require approximately twice the amount of memory. Furthermore, OIL maintains the state of teacher and student models, resulting in a significant memory requirement. This renders OIL almost impractical for real-world applications. Thus, the memory efficiency of Anti-CF makes it a promising choice for deployment, especially in resource-constrained scenarios.

### 5.5 Applications in Generative QA

With the recent amazing progress in generative large language models (LLMs), generative QA is becoming increasingly valuable for research and application. In light of this, we also investigate the

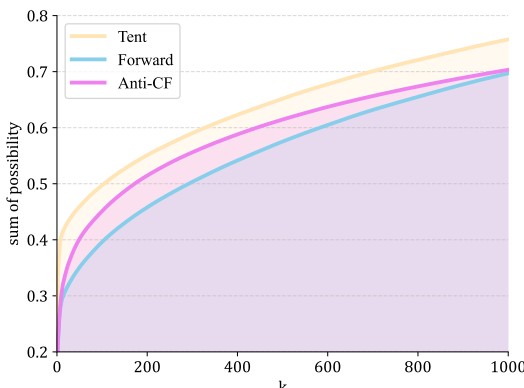

Figure 8: Probability distribution of tokens generated in BLOOM. The ordinate represents the sum of the probabilities of generating the top-$k$ tokens.

| Model | NoiseQA-syn | NoiseQA-na |
|-------|-------------|------------|
| Bloom | 48.77 | 48.12 |
| Tent | 49.08 | 48.85 |
| Ours | 50.70 | 49.83 |

Table 2: EM on NoiseQA.

potential of TTA, especially Anti-CF in it.

We train BLOOM-1.7b (Scao et al., 2022) on the SQuAD dataset using a generative QA setup. Firstly, we study the distribution of generated tokens in generative QA. We employ three methods during inference: direct inference, Tent, and Anti-CF. For each method, we record the probability of each token appearing throughout the inference process. We then sum the probabilities of the top-$k$ tokens. The results of this analysis are presented in Figure 8. The analysis reveals the presence of an imbalanced label distribution in generative QA. High-frequency words are easier to be generated by LLMs. Moreover, Tent will still exacerbate the imbalance. However, by utilizing Anti-CF, this problem can be alleviated to some extent. Anti-CF helps mitigate the imbalanced label distribution's effects and promotes a more balanced generation of words during the inference process. Table 2 shows the EM score of each method on NoiseQA. From the results, we can see that the effect of Anti-CF is better than that of Tent. TTA can still play a role in generative LLMs, and applying TTA in LLMs is a feasible path with great potential.

## 6 Related Work

**Test-Time Adaptation** Test-Time Adaptation (TTA) is a promising paradigm to deal with the distribution shift. TTA uses self-supervised signals to update the model at the inference stage. It has achieved surprising performance in various tasks (Liang et al., 2023). Depending on whether or not modifying the objectives of the training phase and accessing the training data, TTA can be divided into test-time training and fully test-time adaptation. Test-time training (Sun et al., 2020; Liu et al., 2021; Bartler et al., 2022; Gandelsman et al., 2022; Chen et al., 2022) is dedicated to designing a distribution shift-aware auxiliary task during the training phase and using this task to update the model during the testing phase. On the other line, the fully test time adaptation (Wang et al., 2021b) updates the model with test data using a self-training strategy without modifying the training process. However, fully test-time adaptation faces performance degradation due to the low quality of pseudo or soft labels, so current works (Niu et al., 2022; Zhang et al., 2022; Jang and Chung, 2022; Gong et al., 2022; Niu et al., 2023; Song et al., 2023) focus on improving the quality of pseudo-labels, designing more robust and efficient tuning methods. In NLP, TTA has also attracted researchers' attention. Ye et al. (2022) first evaluate the effectiveness of fully test-time adaptation in the QA task and introduce an imitation learning paradigm to ensure the quality of pseudo-labels.

**Robustness NLP** Training a sufficiently robust model is a prerequisite for the practical application of a trustworthy NLP system. Wang et al. (2022) divide robustness in NLP into two categories: adversarial robustness under artificial attacks and naturally occurring distribution shifts. For adversarial robustness, researchers (Ebrahimi et al., 2018; Li et al., 2018; Alzantot et al., 2018; Jia et al., 2019; Garg and Ramakrishnan, 2020; Li et al., 2020; Lin et al., 2021; Ravichander et al., 2021) have proposed many methods for constructing adversarial attack samples to evaluate the model's adversarial robustness. In NLP, naturally occurring distribution shifts also have rich real-world scenarios, such as cross-lingual (Hu et al., 2020; Jafari et al., 2021; Zheng et al., 2021; Yang et al., 2022; Mao et al., 2022), cross-domain (Hendrycks et al., 2020; Ramponi and Plank, 2020; Malinin et al., 2021), and on different styles of text. Many robustness tuning methods are proposed to improve the robustness of NLP models, such as data augmentation (Kaushik et al., 2020; Khashabi et al., 2020; Chen et al., 2020, 2021b; Zhou et al., 2021) and adversarial training

([Miyato et al., 2017](); [Madry et al., 2018](); [Zhu et al., 2020](); [Wang et al., 2021a]()).

## 7 Conclusion

In this paper, we attempt to improve the robustness of QA models by testing time adaptation (TTA) but find that TTA causes the models collapse. We thoroughly investigate why previous TTA methods cause the model collapse and find that the imbalanced label distribution is the main reason. We address this problem by adding constraints between the source and adapted model during the TTA process. We also design an efficient side block to speed up the inference time. Sufficient experimental results show that our proposed method is effective and efficient, making TTA a big step closer to being applied in real-world scenarios.

## Limitations

Although our proposed Anti-CF has made significant progress in terms of stability and inference efficiency compared with existing TTA methods, there are still some limitations:

- Anti-CF constrains the adapted model's prediction with the source model's prediction, which prevents TTA from model collapse and effectively improves the lower bound of the model performance. However, the source model tends to perform poorly under distribution shift, and a strong constraint similar to KL divergence can limit the upper bound of the model performance. The experimental results in Table 1 also imply this limitation. Therefore, we will explore more flexible constraints in the future.
- In this paper, we empirically design the efficient side block without exploring the impact of its structure and capacity on the performance of TTA. The side block still has great potential to be explored, and we hope that future work will focus on this aspect.
- In this paper, we only validate the effectiveness of Anti-CF in extractive and generative QA. However, Anti-CF is task-independent and model-independent, so we believe that Anti-CF is also applicable to other NLP tasks and pre-trained language models. But we do not verify it because of time and computing power constraints. We will further validate the generalizability of Anti-CF in the future.

## Acknowledgements

This work is supported by the National Science Foundation of China (NSFC No. 62206194), the Natural Science Foundation of Jiangsu Province, China (Grant No. BK20220488).

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

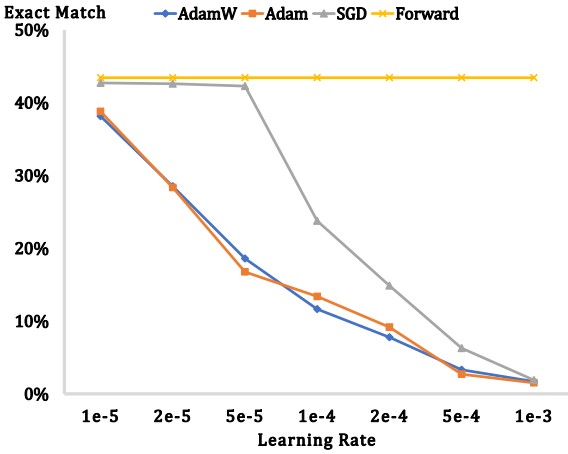

Figure 9: EM (%) on NaturalQA of Forward and Tent under different optimizers and learning rates.

Chen Zhu, Yu Cheng, Zhe Gan, Siqi Sun, Tom Goldstein, and Jingjing Liu. 2020. Freelb: Enhanced adversarial training for natural language understanding. In *International Conference on Learning Representations*.

## A   The Impact of Different Optimizers and Learning Rates on Tent

## B   Hyper parameters

Table 3 shows the detail of hyper parameters.

| Methods | xlmr-base | xTune | xlmr-large |
|---------|-----------|-------|------------|
| Tent | LR: 5e-5 | LR: 5e-5 | LR: 5e-5 |
| EATA | LR: 5e-5
$\beta$: 1/2000
$E_0$: $0.4ln(512)$ | LR: 5e-5
$\beta$: 1/2000
$E_0$: $0.4ln(512)$ | LR: 5e-5
$\beta$: 1/2000
$E_0$: $0.4ln(512)$ |
| OIL | LR: 1e-4
K: 1
$\gamma$: 0.5
$\alpha$: 0.99
$\beta$: 1 | LR: 1e-4
K: 1
$\gamma$: 0.5
$\alpha$: 0.99
$\beta$: 1 | LR: 5e-4
K: 1
$\gamma$: 0.5
$\alpha$: 0.99
$\beta$: 1 |
| SAR | LR: 1e-4
$E_m$: 0.9
$\rho$: 0.05
$E_0$: $0.4ln(512)$ | LR: 1e-4
$E_m$: 0.9
$\rho$: 0.05
$E_0$: $0.4ln(512)$ | LR: 5e-4
$E_m$: 0.9
$\rho$: 0.05
$E_0$: $0.4ln(512)$ |
| Ours | LR: 5e-5
$\alpha$: 0.2 | LR: 5e-5
$\alpha$: 0.2 | LR: 5e-5
$\alpha$: 0.01 |

Table 3:  Details of hyper parameters.

## C   Other start and end positions

Figure 10 shows the start positions of ground truth on NaturalQA and Figure 11 shows the end positions.

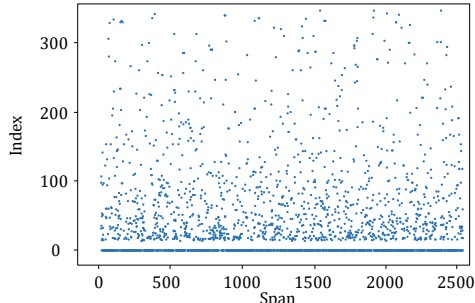

Figure 10: Ground truth of start positions on NaturalQA.

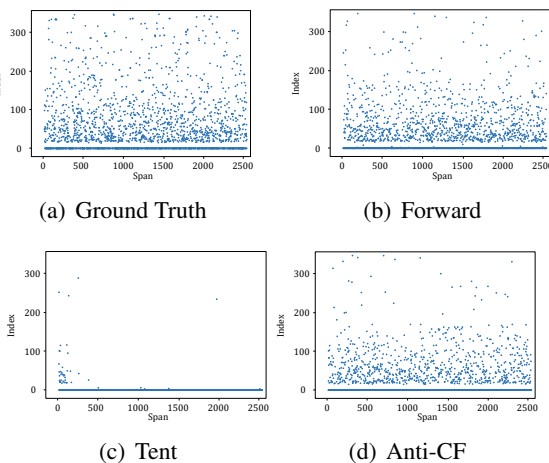

(a) Ground Truth

(b) Forward

(c) Tent

(d) Anti-CF

Figure 11: The end positions of NaturalQA given by the model.

## D   Further analysis on Named Entity Recognition (NER)

In cases where the label distribution in the test dataset is extremely imbalanced, most TTA methods can lead to model collapse, and Anti-CF can effectively avoid this situation. Based on this idea, we design the following experiment.

We have extended Anti-CF to NER. We use RockNER (Lin et al., 2021) to conduct our experiment , which is generated from entity-level adversarial attacks on the OntoNotes (Weischedel et al., 2013) dataset. We want to demonstrate that Anti-CF can effectively improve model performance on RockNER and will not collapse in extreme situa-

| Methods | RockNER | Corrupted | Time |
|---------|---------|-----------|------|
| BERT-base | 56.9 | 56.9 | 0.51 |
| +Tent | 60.6 | 10.7 | 1.71 |
| +EATA | 60.5 | 15.7 | 1.80 |
| +OIL | 58.3 | 56.8 | 4.25 |
| +Ours | 60.2 | 58.0 | 0.78 |

Table 4:  Results on NER.

tions. To prove the second point, we artificially corrupted RockNER by placing samples with more 'O' in front of it. The model will first infer and update samples labelled with more 'O', making the model easier to collapse due to this extreme label distribution. The results are shown in Table 4.