# OpenReview forum: "Beware of Model Collapse! Fast and Stable Test-time Adaptation for Robust Question Answering"
_EMNLP/2023/Conference — EMNLP 2023 Main_

### Official Review · Reviewer_76Fj · 2023-08-04

**Soundness:** 3

**Excitement:**

2: Mediocre: This paper makes marginal contributions (vs non-contemporaneous work), so I would rather not see it in the conference.

**Missing References:**

Hu et al., 2022, LoRA: Low-Rank Adaptation of Large Language Models. ICLR. https://openreview.net/forum?id=nZeVKeeFYf9

**Paper Topic And Main Contributions:**

This paper proposes Anti-Collapse Fast (Anti-CF) test-time adaptation (TTA) for extractive question answering. TTA methods (in particular TENT (Wang et al., 2021)) are not stable and may cause model collapse (i.e., producing almost all incorrect predictions at test time). The paper presents a regularization method that computes KL divergence between the predicted probability produced by the adapted model and the one made by the source model. In addition, only small parameters on the so-called side block are tuned to reduce the parameter size and inference time. The results on various QA datasets indicate that Anti-CF performs on par or better than other models regarding EM/F1 scores but is faster than all adaptive models.

Comments
- In Section 2.2, the paper discusses why the performance of Tent (Wang et al., 2021) significantly drops when adapted to the NaturalQA datasets. However, this issue can also be caused by various factors. In Tent's paper, they used different optimizers, batch sizes, and learning rates for ImageNet and other datasets. I am unsure whether this paper had experimented with a sufficient range of hyperparameters before concluding that Tent leads to model collapse.

- In Eq. (2), incorporating KL divergence between two different predicted probabilities is similar to that of X-MIXUP (the last term of Eq. (7) in Yang et al. (2021)). Although this paper computes the loss at test time, it is worth mentioning/citing it.

- Eq. (4) can be thought of as a special case (i.e., test-time case) of X-MIXUP's loss (Eq. (8)), where L_task is replaced with L_s and the MSE term is omitted.

- The idea of the efficient side block is closely related to Parameter-Efficient Fine-Tuning (PEFT) (e.g., LoRA (Hu et al., 2021)). Discussing the pros/cons between the efficient side block and those existing techniques would make the paper more solid.

- In Table 1, reporting inference time is helpful. It would be nicer if the paper also shows the total number of parameters for each model.

- In Table 3, the learning rate (LR) choices are strange. Typically, the same LR does not work well for different pre-trained LMs (e.g., XLM-RoBERTa-base vs. XLM-RoBERTa-large). This would lead to suboptimal performance.

References:
- Wang et al., 2021, Tent: Fully Test-Time Adaptation by Entropy Minimization, ICLR. https://openreview.net/forum?id=uXl3bZLkr3c
- Yang et al., 2022, Enhancing Cross-lingual Transfer by Manifold Mixup, ICLR. https://openreview.net/forum?id=OjPmfr9GkVv
- Hu et al., 2022, LoRA: Low-Rank Adaptation of Large Language Models. ICLR. https://openreview.net/forum?id=nZeVKeeFYf9
- Zheng et al., 2021, Consistency Regularization for Cross-Lingual Fine-Tuning, ACL. https://aclanthology.org/2021.acl-long.264/




**Questions For The Authors:**

- Lines 182-184 state that "Therefore, the imbalanced distribution of test labels has led to TTA leaning too much towards the majority class during the update process". Could you clarify the meaning of "labels" and "the majority class" here?


- When computing the predicted probability for the adapted model, do the efficient side block's parameters require forward propagation? If so, lines 237-239 "... as the adapted model so that we only need one forward propagation to obtain the two outputs simultaneously" are incorrect.

**Reasons To Accept:**

The proposed Anti-CF model sounds reasonable and would be applicable to other tasks.

**Reasons To Reject:**

This paper pointed out the problem of Tent (Wang et al., 2021), which can lead to model collapse. However, whether Tent was tuned properly in Section 2.2 is still questionable because Tent can be sensitive to hyperparameters (e.g., learning rate, batch size, etc.) and the optimizer. The results in Table 1 also show that Tent performs reasonably well on other datasets. The authors are encouraged to discuss the proposed side block compared to existing PEFT methods (e.g., LoRA) to justify its novelty.

**Reproducibility:**

3: Could reproduce the results with some difficulty. The settings of parameters are underspecified or subjectively determined; the training/evaluation data are not widely available.

**Reviewer Confidence:**

4: Quite sure. I tried to check the important points carefully. It's unlikely, though conceivable, that I missed something that should affect my ratings.

---

> ### Author Rebuttal · Authors · 2023-08-29
>
> Thanks for your valuable comments and constructive suggestions.  We hope the information attached below can address your concern.
>
>
>
> Q1:
>
> Whether Tent was tuned properly in Section 2.2 is still questionable.
>
> A1:
>
> In fact, we have made a reasonable parameter selection for Tent.
>
> In Section 2.2, We used the AdamW optimizer and selected a learning rate of 1e-4. This is because such settings perform well in most datasets, such as NoiseQA-syn. We have list the performance of Tent in NoiseQA-syn below for reference.
>
> | LR         | 1e-5  | 2e-5  | 5e-5  | 1e-4  | 2e-4  | 5e-4  | 1e-3  |
> | ---------- | ----- | ----- | ----- | ----- | ----- | ----- | ----- |
> | xlmr-base  | 66.61 | 66.61 | 66.61 | 66.61 | 66.61 | 66.61 | 66.61 |
> | Tent+AdamW | 67.11 | 67.51 | 68.14 | 68.52 | 68.54 | 67.87 | 66.11 |
> | Tent+Adam  | 67.20 | 67.54 | 68.07 | 68.54 | 68.49 | 67.90 | 65.85 |
> | Tent+SGD   | 66.69 | 66.67 | 66.67 | 66.64 | 66.95 | 67.11 | 67.42 |
>
>
>
> We also repeat the experiments in the preliminary under different hyperparameters. For batch size, as SAR (Niu et al., 2023) discussed it in detail, BN layers make Tent sensitive to batch size. However, in Transformer based models, there is no BN layer, only LN layers. Therefore, Tent is not sensitive to batch size in our experiment, and our previous practical process has proven this.
>
> We have made a more detailed selection of learning rates and optimizers. All other experimental settings are the same as in Section 2.2. We report the EM values due to space limitations.
>
> | LR         | 1e-5  | 2e-5  | 5e-5  | 1e-4  | 2e-4  | 5e-4  | 1e-3  |
> | ---------- | ----- | ----- | ----- | ----- | ----- | ----- | ----- |
> | xlmr-base  | 43.44 | 43.44 | 43.44 | 43.44 | 43.44 | 43.44 | 43.44 |
> | Tent+AdamW | 38.15 | 28.53 | 18.59 | 11.65 | 7.78  | 3.32  | 1.70  |
> | Tent+Adam  | 38.80 | 28.36 | 16.76 | 13.39 | 9.17  | 2.71  | 1.52  |
> | Tent+SGD   | 42.72 | 42.61 | 42.28 | 23.78 | 14.86 | 6.28  | 1.89  |
>
> Obviously, regardless of learning rates and optimizers, Tent cannot perform well on the NaturalQA dataset. And the trend is that the higher the learning rate, the lower its accuracy, which is also in line with expectations. When the learning rate is set to 0, it is obvious that the model will never collapse, and as the learning rate increases, the model collapses faster.
>
> It should be noted that the goal of TTA is to take effect in real-world application scenarios, and TTA does not make any assumptions about the distribution of test samples. Therefore, it is unreasonable to choose different hyperparameters for different datasets, as we do not know how TTA performs in the test set. In addition, if we can choose different hyperparameters for different datasets, then TTA will never cause the model collapse, because if we have found that TTA causes the model collapse, we can directly set the learning rate to 0, which is a type of cheating.  LAME (Malik et al., 2022) also proposed the same viewpoint.
>
>
>
>
>
> Q2：
>
> The authors are encouraged to discuss the proposed side block compared to existing PEFT methods (e.g., LoRA) to justify its novelty.
>
> A2：
>
> Side block can be considered as a type of PEFT, and its advantage is that it can simultaneously obtain source output and adapted output. However, other PEFT methods such as LoRA cannot achieve this, as they require at least two forward propagation to get two different outputs. In addition, the gradient of Anti-CF only propagates in the side block and no propagation will occur in the backbone, which greatly reduces the time and memory required for backpropagation.
>
> In fact, the idea that we only adjust LayerNorm comes from PEFT, which can also be seen as a simple PEFT, similar to BitFit.
>
> We conduct the entire experiment with Tent again using LoRA. We repeat the experiment three times under different random seeds and report its mean value. Due to space limitations, we only report EM values. The rank we use in LoRA is 4.
>
> | Models     | NoiseQA-syn | NoiseQA-na | XQuAD | MLQA  | MRQA  | Avg.  | Time  |
> | ---------- | ----------- | ---------- | ----- | ----- | ----- | ----- | ----- |
> | xlmr-base  | 68.63       | 67.54      | 57.09 | 48.15 | 16.28 | 51.54 | 10.97 |
> | xTune      | 71.54       | 70.25      | 59.55 | 50.06 | 23.21 | 54.92 | 10.97 |
> | xlmr-large | 70.51       | 68.70      | 63.61 | 52.51 | 7.62  | 52.59 | 33.66 |
>
> We can see that the results of Tent obtained using LoRA are similar to those in our main experiment in terms of accuracy and inference speed.
>
> In addition, we also try to use both LoRA and KL constraints simultaneously. We make two copies of the source model, one frozen and the other updated using LoRA. These two models are called the source model and the adapted model respectively. We obtain the source output through a forward propagation in the source model the adapted output through a forward propagation in the adapted model, then update the adapted model using Eq. (4) in out paper. Obviously, this method requires two forward propagation and one backpropagation in the backbone. The results of the experiment are shown in the table below. We only report EM values due to space limitations.
>
> | Models             | NoiseQA-syn | NoiseQA-na | XQuAD | MLQA  | MRQA  | Avg.  | Time  |
> | ------------------ | ----------- | ---------- | ----- | ----- | ----- | ----- | ----- |
> | xlmr-base+LoRA+KL  | 68.26       | 67.55      | 56.75 | 48.42 | 40.27 | 56.27 | 15.12 |
> | xlmr-base+Anti-CF  | 68.08       | 67.50      | 56.75 | 48.32 | 40.16 | 56.16 | 5.23  |
> | xTune+LoRA+KL      | 71.85       | 71.20      | 59.76 | 51.17 | 43.16 | 59.43 | 15.12 |
> | xTune+Anti-CF      | 71.32       | 70.19      | 59.01 | 51.18 | 43.44 | 59.02 | 5.23  |
> | xlmr-large+LoRA+KL | 70.81       | 68.57      | 63.88 | 54.26 | 46.91 | 60.28 | 45.84 |
> | xlmr-large+Anti-CF | 71.53       | 69.28      | 63.78 | 54.93 | 46.24 | 61.15 | 15.36 |
>
> The results in the table indicate that this approach may perform slightly better than Anti-CF, but it will be about three times slower than Anti-CF. We believe that it is not worth trading such inference time costs for an extremely limited increase in performance. In fact, we have tried this method before, but for faster inference speed, we designed the side block.
>
>
>
>
>
> Q3:
>
> Lines 182-184 state that "Therefore, the imbalanced distribution of test labels has led to TTA leaning too much towards the majority class during the update process". Could you clarify the meaning of "labels" and "the majority class" here?
>
> A3:
>
> Labels refer to the answer position of the current span. When each sample is cut into many spans, the answer is likely not in the current span, so the position defaults to 0. So, many span labels are 0, which is what I call the major class.
>
>
>
> Q4:
>
> When computing the predicted probability for the adapted model, do the efficient side block's parameters require forward propagation? If so, lines 237-239 "... as the adapted model so that we only need one forward propagation to obtain the two outputs simultaneously" are incorrect.
>
> A4:
>
> There will be forward propagation in the side block, but the side block is relatively small compared to the backbone and can generally be ignored. Our side block is similar to a PEFT method, just like LoRA with an additional classifier. We generally cannot say that it obtains output with forward propagation twice. It is incorrect to consider its forward propagation in the backbone and in the low rank matrix as two forward propagation.
>
>
>
> Q5:
>
> In Eq. (2), incorporating KL divergence between two different predicted probabilities is similar to that of X-MIXUP (the last term of Eq. (7) in Yang et al. (2021)). Although this paper computes the loss at test time, it is worth mentioning/citing it.
>
> A5:
>
> Adding KL divergence constraints to normal training is a common idea, so we may have overlooked this paper. We will add this citation to the final version of our paper.
>
>
>
> Q6:
>
> In Table 1, reporting inference time is helpful. It would be nicer if the paper also shows the total number of parameters for each model.
>
> A6:
>
> We have reported the usage of memory in Section 5.4, which can reflect the relative size of parameter quantities. The final version of our paper will show the total number of parameters of each model in Table 1.
>
>
>
>
>
> Q7:
>
> In Table 3, the learning rate (LR) choices are strange. Typically, the same LR does not work well for different pre-trained LMs (e.g., XLM-RoBERTa-base vs. XLM-RoBERTa-large). This would lead to suboptimal performance.
>
> A7:
>
> We have briefly explained the principles for selecting hyperparameters under our settings in lines 336-340. We will provide a more detailed explanation here.
>
> The goal of TTA is to take effect in real-world application scenarios, and TTA does not make any assumptions about the distribution of test samples. Therefore, it is unreasonable to choose different hyperparameters for different datasets, as we do not know how TTA performs in the test set. In addition, if we can choose different hyperparameters for different datasets, then TTA will never cause the model collapse, because if we have found that TTA causes the model collapse, we can directly set the learning rate to 0, which is a type of cheating.  LAME (Malik et al., 2022) also proposed the same viewpoint.
>
> It is reasonable to choose different hyperparameters for different models, because we can use experience to determine which models require a higher learning rate and which require a lower learning rate, which is feasible in real-world applications. So, our setting is: once a model and a TTA method are determined, their hyperparameters cannot be changed.
>
> In our experiment, we have explored a learning rate range of {5e-5, 1e-4, 5e-4}, which can make all TTA methods in our experiment effective. However, in the table, Tent and EATA tend to choose 5e-5 because model collapse can be alleviated at low learning rates. Regarding hyperparameters beyond the learning rate, we have respectively referred to the hyperparameter settings of OIL and EATA.
>
> It should be noted that Tent and EATA still perform well on other datasets under 5e-5, so there is no suspicion that we intentionally lowered the baseline. We report the EM values of Tent under different learning rates and datasets under xlmr-base.
>
> | LR   | NoiseQA-syn | NoiseQA-na | XQuAD | MLQA  | MRQA  | Avg.  |
> | ---- | ----------- | ---------- | ----- | ----- | ----- | ----- |
> | 5e-5 | 68.14       | 67.44      | 56.40 | 48.11 | 20.77 | 52.17 |
> | 1e-4 | 68.40       | 67.67      | 56.78 | 48.16 | 16.52 | 51.51 |
> | 5e-4 | 67.12       | 66.79      | 56.50 | 47.70 | 6.34  | 48.90 |
>
> In addition, more precise selection of hyperparameters can also improve the performance of Anti-CF. We report the EM values of Anti-CF under xlmr-large and NoiseQA-syn with different hyperparameters.
>
> | LR/$\alpha$ | 0.01  | 0.2   | 0.8   |
> | ----------- | ----- | ----- | ----- |
> | 5e-5        | 68.21 | 68.08 | 66.92 |
> | 1e-4        | 67.51 | 67.82 | 66.86 |
> | 5e-4        | 65.61 | 67.00 | 66.65 |
>
>
>
>
>
> References:
>
> - Niu et al., 2023, Towards stable test-time adaptation in dynamic wild world, ICLR
> - Malik et al., 2022, Parameter-free Online Test-time Adaptation, CVPR.

---

### Official Review · Reviewer_CXeg · 2023-08-06

**Soundness:** 4

**Excitement:**

4: Strong: This paper deepens the understanding of some phenomenon or lowers the barriers to an existing research direction.

**Paper Topic And Main Contributions:**

This paper aims to improve the application of pre-trained language models (PLMs) on tasks/scenarios where a distribution shift is expected. It has been observed that the performance of PLMs deteriorates when the test distribution does not match the training distribution. In literature, some attempts have been made to update the model during the testing phase to make the model adapted for distribution shift. However, due to the unreliable output, the test time adaptation (TTA) strategy also fails. This study contributes in the following ways -
1. Consider a particular task of Question-Answering and investigates why TTA causes model collapse
2. Proposes Anti-Collapse Fast test-time adaptation (Anti-CF) to mitigate model collapse issue. Anti-CF treats outputs of the source model as a soft label to regularize model updates during test time.
3. Extensive empirical studies to show the proposed Anti-CF can effectively take care of model collapse.


**Questions For The Authors:**

Anti-CF with xlmr-base and xTune is underperforming compared to the other baseline models for NoiseQA-Syn, NoiseQA-na and XSQuAD tasks, whereas it is performing well for MLQA and MRQA. Is there any specific explanation for this?

**Reasons To Accept:**

1. This work proposes an approach to prevent model collapse during test time adaptation.
2. By introducing a side-block along with the main model, it has significantly reduced the computational cost and presents an efficient approach
3. Experiments are carefully designed and presented nicely

**Reasons To Reject:**

Although this study shows significant improvements in TTA for extractive QA task, it did not show the performance measure of any other task. Thus, it is not clear about the generalizability of the proposed method for other tasks. Offcourse, it is hard to include several tasks within the given space, some discussions along the line would have been great.

**Reproducibility:**

4: Could mostly reproduce the results, but there may be some variation because of sample variance or minor variations in their interpretation of the protocol or method.

**Reviewer Confidence:**

3: Pretty sure, but there's a chance I missed something. Although I have a good feel for this area in general, I did not carefully check the paper's details, e.g., the math, experimental design, or novelty.

---

> ### Author Rebuttal · Authors · 2023-08-29
>
> Thanks for your comments and acknowledgment. We hope the following responses can address your concern:
>
>
>
> Q1:
>
> Although this study shows significant improvements in TTA for extractive QA task, it did not show the performance measure of any other task.  Thus, it is not clear about the generalizability of the proposed method for other tasks. Offcourse, it is hard to include several tasks within the given space, some discussions along the line would have been great.
>
> A1:
>
> Anti-CF can achieve comparable or better performance in most scenarios where TTA performs well. In addition, in cases where the label distribution in the test dataset is extremely imbalanced, most TTA methods can lead to model collapse, and Anti-CF can effectively avoid this situation. Based on this idea, we design the following experiment.
>
> We have extended Anti-CF to NER. We use RockNER (Lin et al., 2021) to conduct our experiment , which is generated from entity-level adversarial attacks on the OntoNotes (Weischedel et al., 2013) dataset. We want to demonstrate that Anti-CF can effectively improve model performance on RockNER and will not collapse in extreme situations. To prove the second point, we artificially corrupted RockNER by placing samples with more 'O' in front of it. The model will first infer and update samples labelled with more 'O', making the model easier to collapse due to this extreme label distribution. The results are shown in the table below. More experimental details will be presented in the final version of our paper.
>
> | methods   | RockNER | Corrupted | Time |
> | --------- | ------- | --------- | ---- |
> | BERT-base | 56.9    | 56.9      | 0.51 |
> | +Tent     | 60.6    | 10.7      | 1.71 |
> | +EATA     | 60.5    | 15.7      | 1.80 |
> | +OIL      | 58.3    | 56.8      | 4.25 |
> | +Ours     | 60.2    | 58.0      | 0.78 |
>
> We can see that Anti-CF can still achieve comparable performance compared to other TTA methods and can prevent the model from collapsing with the fastest inference speed.
>
>
>
>
>
> Q2:
>
> Anti-CF with xlmr-base and xTune is underperforming compared to the other baseline models for NoiseQA-Syn, NoiseQA-na and XSQuAD tasks, whereas it is performing well for MLQA and MRQA. Is there any specific explanation for this?
>
> A2:
>
> Even though Anti-CF can effectively prevent model collapse, it still limits the upper bound of performance to some extent, as we  have discussed in the Limitation section of our paper. So on datasets such as NoiseQA-syn, NoiseQA-na, and XQuAD that already perform well with TTA, the performance of Anti-CF may be slightly lower. On datasets with poor TTA performance such as MLQA and MRQA, Anti CF performs better than other methods.
>
> From another perspective, MLQA and MRQA have longer sentence lengths, resulting in more segmentation during preprocessing, resulting in more labels falling at position 0 (indicating the answer is not in the current span). Therefore, other TTA methods perform mediocrely on datasets such as MLQA and MRQA, and this is where Anti-CF can come into play.
>
>
>
>
>
> References:
>
> - Lin et al., 2021, A simple method to create adversarial examples for evaluating the robustness of named entity recognition models, EMNLP.
> - Weischedel et al., 2013, Ontonotes release 5.0 ldc2013t19, Linguistic Data Consortium.

---

### Official Review · Reviewer_kMrG · 2023-08-10

**Soundness:** 4

**Excitement:**

4: Strong: This paper deepens the understanding of some phenomenon or lowers the barriers to an existing research direction.

**Paper Topic And Main Contributions:**

This paper proposes a new TTA method called Anti-CF to address the problem of performance degradation of pre-trained language models in distribution shift scenarios. It discusses the causes of potential model collapse issues that could be caused by TTA, and improves the stability and efficiency of TTA in practical applications.

**Questions For The Authors:**

PCL NB

**Reasons To Accept:**

1. By analyzing, it was discovered that the imbalance in label distribution in the test set of the question answering task is a major cause of model collapse in TTA methods. This finding is of great significance and provides important guidance.
2. The Anti-CF method is proposed, which uses the output of the source model as soft label constraints for the adapted model in the TTA process, effectively preventing model collapse.
3. Anti-CF is demonstrated to achieve stable performance improvements in various distribution shift scenarios, making TTA more reliable in practical applications.

**Reasons To Reject:**

1. Anti-CF uses KL divergence to constrain the adapted model's output to not deviate too much from the source model, which limits the upper bound of performance to some extent. More flexible constraint methods could be explored.
2. Could the author compare with stronger TTA methods? For example, SAR (https://openreview.net/pdf?id=g2YraF75Tj), AdaNPC (https://arxiv.org/abs/2304.12566), LAME (https://arxiv.org/abs/2201.05718).
3. Anti-CF is currently validated only in the question answering task, and it can be extended to other NLP tasks in the future.
4. The selection of hyperparameters in the experiments could be more systematic. It needs further investigation whether different datasets and models require different combinations of hyperparameters. The computational and memory complexity of Anti-CF can be analyzed in more detail.

**Reproducibility:**

4: Could mostly reproduce the results, but there may be some variation because of sample variance or minor variations in their interpretation of the protocol or method.

**Reviewer Confidence:**

3: Pretty sure, but there's a chance I missed something. Although I have a good feel for this area in general, I did not carefully check the paper's details, e.g., the math, experimental design, or novelty.

---

> ### Author Rebuttal · Authors · 2023-08-29
>
> Thanks a lot for your valuable comments. We hope the following responses can address your concern:
>
> Q1:
>
> Anti-CF uses KL divergence to constrain the adapted model's output to not deviate too much from the source model, which limits the upper bound of performance to some extent. More flexible constraint methods could be explored.
>
> A1:
>
> This is correct, and we have also indicated this issue in our Limitation. In fact, it is possible to have better constraints that enable the model to avoid collapse while not limiting the upper bound of performance. But we want to say that KL divergence is already a very good constraint, so considering our limited time and computing resources, we did not explore other constraints.
>
>
>
> Q2:
>
> Could the author compare with stronger TTA methods? For example, SAR , AdaNPC, LAME.
>
> A2:
>
> Thank you for your suggestion. In fact, most of the existing TTA methods are only validated in CV tasks, and we have included existing baselines in the NLP field. Nevertheless, to address your concerns, we will still migrate some baselines in the CV field to NLP. We have added two new baselines, SAR and LAME (we don't select AdaNPC because it requires retraining the classification head with KNN, which may result in an unfair comparison). All experimental settings are the same as the other baselines in the paper. We repeat our experiments three times under different random seeds and report their mean values. We will add these two baselines to the final version of our paper. Due to space limitations, we only report the EM values here.
>
>
>
> | Models     | NoiseQA-syn | NoiseQA-na | XQuAD | MLQA  | MRQA  | Avg.  | Time  |
> | ---------- | ----------- | ---------- | ----- | ----- | ----- | ----- | ----- |
> | xlmr-base  | 66.61       | 66.08      | 55.59 | 48.27 | 40.12 | 55.33 | 4.25  |
> | +SAR       | 67.63       | 67.36      | 55.94 | 48.01 | 31.71 | 54.13 | 18.30 |
> | +LAME      | 66.43       | 65.88      | 55.47 | 48.10 | 39.20 | 55.02 | 4.62  |
> | +Ours      | 68.08       | 67.50      | 56.75 | 48.32 | 40.16 | 56.16 | 5.23  |
> | xTune      | 70.92       | 69.72      | 58.81 | 50.66 | 43.46 | 58.70 | 4.25  |
> | +SAR       | 70.90       | 70.34      | 59.23 | 51.19 | 36.73 | 57.68 | 18.30 |
> | +LAME      | 70.64       | 69.40      | 58.52 | 50.32 | 42.46 | 58.27 | 4.62  |
> | +Ours      | 71.32       | 70.19      | 59.01 | 51.18 | 43.44 | 59.02 | 5.23  |
> | xlmr-large | 65.55       | 64.17      | 63.15 | 54.40 | 46.17 | 58.69 | 12.37 |
> | +SAR       | 69.29       | 67.70      | 63.52 | 54.43 | 46.41 | 60.27 | 52.47 |
> | +LAME      | 65.35       | 63.94      | 62.97 | 54.12 | 45.35 | 58.34 | 12.63 |
> | +Ours      | 71.53       | 69.28      | 63.78 | 54.93 | 46.24 | 61.15 | 15.36 |
>
>
>
> From the table, we can see that Anti-CF still outperforms these two stronger baselines.
>
> The performance of SAR is stronger than the other baselines selected in our paper. It searches for the optimal solution within the neighborhood, making the performance of TTA more stable. Therefore, it can alleviate the problem of model collapse to some extent. But due to its need for two forward propagation and two backward propagation, its inference time is approximately twice that of Tent.
>
> LAME does not update the parameters of the model, only adjust the output probability. Although this can avoid model collapse, it also greatly limits the performance of TTA. In addition, due to the differences between CV tasks and NLP tasks, LAME has almost no effect on our datasets.
>
>
>
>
>
> Q3:
>
> Anti-CF is currently validated only in the question answering task, and it can be extended to other NLP tasks in the future.
>
> A3:
>
> Anti-CF can achieve comparable or better performance in most scenarios where TTA performs well. In addition, in cases where the label distribution in the test dataset is extremely imbalanced, most TTA methods can lead to model collapse, and Anti-CF can effectively avoid this situation. Based on this idea, we design the following experiment.
>
> We have extended Anti-CF to NER. We use RockNER (Lin et al., 2021) to conduct our experiment , which is generated from entity-level adversarial attacks on the OntoNotes (Weischedel et al., 2013) dataset. We want to demonstrate that Anti-CF can effectively improve model performance on RockNER and will not collapse in extreme situations. To prove the second point, we artificially corrupted RockNER by placing samples with more 'O' in front of it. The model will first infer and update samples labelled with more 'O', making the model easier to collapse due to this extreme label distribution. The results are shown in the table below. More experimental details will be presented in the final version of our paper.
>
> | methods   | RockNER | Corrupted | Time |
> | --------- | ------- | --------- | ---- |
> | BERT-base | 56.9    | 56.9      | 0.51 |
> | +Tent     | 60.6    | 10.7      | 1.71 |
> | +EATA     | 60.5    | 15.7      | 1.80 |
> | +OIL      | 58.3    | 56.8      | 4.25 |
> | +Ours     | 60.2    | 58.0      | 0.78 |
>
> We can see that Anti-CF can still achieve comparable performance compared to other TTA methods and can prevent the model from collapsing with the fastest inference speed.
>
>
>
> Q4:
>
> The selection of hyperparameters in the experiments could be more systematic. It needs further investigation whether different datasets and models require different combinations of hyperparameters.
>
> A4:
>
> We have briefly explained the principles for selecting hyperparameters under our settings in lines 336-340. We will provide a more detailed explanation here.
>
> The goal of TTA is to take effect in real-world application scenarios, and TTA does not make any assumptions about the distribution of test samples. Therefore, it is unreasonable to choose different hyperparameters for different datasets, as we do not know how TTA performs in the test set. In addition, if we can choose different hyperparameters for different datasets, then TTA will never cause the model collapse, because if we have found that TTA causes the model collapse, we can directly set the learning rate to 0, which is a type of cheating.  LAME (Malik et al., 2022) also proposed the same viewpoint.
>
> It is reasonable to choose different hyperparameters for different models, because we can use experience to determine which models require a higher learning rate and which require a lower learning rate, which is feasible in real-world applications. So, our setting is: once a model and a TTA method are determined, their hyperparameters cannot be changed.
>
> In our experiment, we have explored a learning rate range of {5e-5, 1e-4, 5e-4}, which can make all TTA methods in our experiment effective. However, in the table, Tent and EATA tend to choose 5e-5 because model collapse can be alleviated at low learning rates. Regarding hyperparameters beyond the learning rate, we have respectively referred to the hyperparameter settings of OIL and EATA.
>
> It should be noted that Tent and EATA still perform well on other datasets under 5e-5, so there is no suspicion that we intentionally lowered the baseline. We report the EM values of Tent under different learning rates and datasets under xlmr-base.
>
> | LR   | NoiseQA-syn | NoiseQA-na | XQuAD | MLQA  | MRQA  | Avg.  |
> | ---- | ----------- | ---------- | ----- | ----- | ----- | ----- |
> | 5e-5 | 68.14       | 67.44      | 56.40 | 48.11 | 20.77 | 52.17 |
> | 1e-4 | 68.40       | 67.67      | 56.78 | 48.16 | 16.52 | 51.51 |
> | 5e-4 | 67.12       | 66.79      | 56.50 | 47.70 | 6.34  | 48.90 |
>
> In addition, more precise selection of hyperparameters can also improve the performance of Anti-CF. We report the EM values of Anti-CF under xlmr-large and NoiseQA-syn with different hyperparameters.
>
> | LR/$\alpha$ | 0.01  | 0.2   | 0.8   |
> | ----------- | ----- | ----- | ----- |
> | 5e-5        | 68.21 | 68.08 | 66.92 |
> | 1e-4        | 67.51 | 67.82 | 66.86 |
> | 5e-4        | 65.61 | 67.00 | 66.65 |
>
>
>
> Q5:
>
> The computational and memory complexity of Anti-CF can be analyzed in more detail.
>
> A5:
>
> We have reported the inference speed of Anti-CF in Section 4.4 and the memory requirements in Section 5.4. We believe that the results in our paper fully demonstrate the superiority of Anti-CF over other TTA methods. Due to time constraints during rebuttal period, we will add further analysis in the final version of our paper.
>
>
>
> References:
>
> - Lin et al., 2021, A simple method to create adversarial examples for evaluating the robustness of named entity recognition models, EMNLP.
> - Weischedel et al., 2013, Ontonotes release 5.0 ldc2013t19, Linguistic Data Consortium.
> - Malik et al., 2022, Parameter-free Online Test-time Adaptation, CVPR.

---

### Meta-Review · Area_Chair_kHuP · 2023-09-25

**Recommendation:** 4

**Metareview:**

The paper proposes a new TTA method called Anti-CF to address the problem of performance degradation of pre-trained language models in distribution shift scenarios. The authors applied their methods to extractive question answering.

Reviewers agree that the approach is reasonable and that could even be applied to other NLP tasks. Some reviewers highlight the importance of the analysis leading to discover that the imbalance in label distribution in the test set of the question answering task is a major cause of model collapse in TTA method. Some reviewers also mention that the approach if efficient, and that the authors have designed a careful set of experiments to demonstrate that their proposed approach achieves stable performance improvements in various distribution shift scenarios.

The main concern raised by one reviewer is related to novelty, mainly because the authors could explain better how their proposed method differs and/or builds on existing approaches (e.g. LAME and PEFT techniques). Some of these clarifications were provided during the rebuttal period, showcasing that the authors are aware of these techniques and their relationship with their work. As such, they are strongly encouraged to include these comparisons and, in turn, adapt their novelty claims. Other concerns were related to comparisons with stronger models, which were presented during rebuttal, as well as details on hyperparameter settings (also added during rebuttal). Finally, there was some concern regarding using KL divergence and the limits it brings to the approach, which the authos have acknowledged is a limitation (it is stated in the corresponding section in the paper) and plan on tackling it in future work.

---

### Decision · Program_Chairs · 2023-10-07

**Decision:**

Accept-Main

**Comment:**

The paper proposes a new TTA method called Anti-CF to address the problem of performance degradation of pre-trained language models in distribution shift scenarios. The authors applied their methods to extractive question answering.

Reviewers agree that the approach is reasonable and that could even be applied to other NLP tasks. Some reviewers highlight the importance of the analysis leading to discover that the imbalance in label distribution in the test set of the question answering task is a major cause of model collapse in TTA method. Some reviewers also mention that the approach if efficient, and that the authors have designed a careful set of experiments to demonstrate that their proposed approach achieves stable performance improvements in various distribution shift scenarios.

The main concern raised by one reviewer is related to novelty, mainly because the authors could explain better how their proposed method differs and/or builds on existing approaches (e.g. LAME and PEFT techniques). Some of these clarifications were provided during the rebuttal period, showcasing that the authors are aware of these techniques and their relationship with their work. As such, they are strongly encouraged to include these comparisons and, in turn, adapt their novelty claims. Other concerns were related to comparisons with stronger models, which were presented during rebuttal, as well as details on hyperparameter settings (also added during rebuttal). Finally, there was some concern regarding using KL divergence and the limits it brings to the approach, which the authos have acknowledged is a limitation (it is stated in the corresponding section in the paper) and plan on tackling it in future work.